# The impact of straw and its post-pyrolysis incorporation on functional microbes and mineralization of organic carbon in yellow paddy soil

**Fangchi Wang**[1], **Xiaoli Wang**[1]*, **Jianjun Duan**[2], **Sanwei Yang**[1], **Jie Wei**[1‡], **Shengmei Yang**[1‡], **Qinwen Zheng**[1‡]

1 College of Agriculture, Guizhou University, Guiyang, China, 2 Key Laboratory of Tobacco Quality Research of Guizhou Province, College of Tobacco, Guizhou University, Guiyang, China

☯ These authors contributed equally to this work.
‡ JW, SY and QZ also contributed equally to this work.
* xlwang@gzu.edu.cn

**Data Availability Statement:** All relevant data are within the article and its Supporting information files.

## Abstract

The impact of straw and biochar on carbon mineralization and the function of carbon cycle genes in paddy soil is important for soil nutrient management and the transformation of carbon pools. This research is based on a five-year field experiment with four treatments: no fertilizer application (CK); chemical fertilizer only (NPK); straw combined with chemical fertilizer (NPKS); and biochar combined with chemical fertilizer (NPKB). By integrating indoor mineralization culture with metagenomic approaches, we analyzed the response of organic carbon mineralization and carbon cycle genes in typical paddy soil from Guizhou Province, China, to different fertilization treatments. The result shows that the various fertilization treatments significantly increased the levels of soil organic carbon, dissolved organic carbon, microbial biomass carbon, and readily oxidizable organic carbon. The NPKS treatment increased the rate of soil organic carbon mineralization, whereas the NPKB treatment decreased it. Overall, the NPK and NPKB treatments increased the relative abundance of carbon fixation genes. The NPKS treatment increased the relative abundance of carbon degradation genes. The NPKS treatment increased the abundance of Proteobacteria, whereas the NPKB treatment decreased the abundance of Actinobacteria. Biochar after straw pyrolysis can reduce carbon loss and enhance sequestration of soil carbon, whereas straw decreases soil organic carbon stability, accelerating the transformation of soil carbon pools. Future research should encompass long-term impact assessments to comprehensively understand the enduring effects of these fertilization treatments on soil carbon mineralization and the function of carbon cycle genes.

## Introduction

Soil represents the largest carbon reservoir within ecosystems, with even minor fluctuations potentially influencing atmospheric $CO_2$ concentrations [1]. Organic carbon is a pivotal

**Funding:** This research was funded by the Construction of High Quality and Efficient Mechanized Scientific and Technological Innovation Talent Team of Characteristic Coarse Cereals in Guizhou Province (grant number BQW [2024]009) and the Research and Integrated Application of Key Technologies of Green and High Yield in Characteristic Mountain Agriculture (grant number [2023]07). The funders had no role in study design, data collection and analysis, decision to publish, or preparation of the manuscript.

component of the soil carbon pool, and its mineralization through microbial decomposition is crucial for altering soil carbon sequestration. Research indicates that soil microorganisms are instrumental in modulating agricultural productivity and the carbon cycle. The structure and function of microbial communities have an important impact on soil carbon emissions, and are also the main driving force of the carbon cycle in paddy soil [2]. Understanding the soil functional microbiome is essential for elucidating ecological processes and functions, including nutrient cycling and the conversion of carbon to carbon dioxide. Variations in the abundance of genes associated with carbon metabolism and shifts in microbial community structure can result in disparities in carbon metabolic functions. These differences, in turn, can affect the equilibrium of soil as both a 'C source' and a 'C sink' within agroecosystems [3]. Consequently, investigation of soil microorganisms offers valuable insights into the regulatory mechanisms of the soil carbon cycle. In the context of global climate warming and the aggressive promotion of straw recycling, examining the impacts of direct straw application and carbonization treatments on the mineralization of soil organic carbon and its associated functional genes is of paramount importance. Such research is not only critical for understanding soil carbon pools and climate change interactions, but also provides a theoretical foundation for informed fertilization strategies.

Agriculture is one of the main sources of global emissions that contribute to climate change [4]. China is accountable for nearly 27% of total global $CO_2$ emissions, making it the largest emitter worldwide [5]. Similarly, China stands as a global powerhouse in terms of straw resources, with an annual output that can reach up to $9.40 \times 10^8$ tons [6]. In the context of the global objectives of achieving a 'carbon peak' and 'carbon neutrality', the practice of straw incorporation, through either direct or pyrolytic methods, holds important practical implications for soil carbon sequestration and fertilization potential. As an important renewable resource, straw contains mineral nutrients and a large amount of organic matter necessary for plant growth, and is an important source of organic fertilizer in soil. However, as a material high in both carbon and nitrogen, the return of straw to the soil can lead to the immobilization of inorganic nitrogen sources by microorganisms, thereby initiating competition for nitrogen between microorganisms and crops. Consequently, the practice of straw incorporation is often paired with nitrogen fertilization to effectively enhance soil organic carbon levels, reduce soil bulk density, facilitate soil aggregate formation, and optimize the thermal and moisture conditions necessary for crop growth [7]. Nevertheless, the incorporation of straw may also accelerate the decomposition of organic carbon, potentially exerting a negative influence on net carbon sequestration. This decreased sequestration is primarily attributed to the increase in microbial biomass and activity, as well as carbon losses through mineralization and methane production in both aerobic and anaerobic soils [8]. The recent findings of Yang et al. [9] suggest that continuous straw incorporation over three years can increase the original soil organic carbon content but also accelerate the organic carbon mineralization rate. Yin et al. [10] found that after 112 days of incubation, biochar application significantly increased soil organic carbon content, whereas straw application increased the active components of soil organic carbon and accelerated the soil organic carbon mineralization rate. Biochar, the product of the thermal degradation of organic matter in an oxygen-limited environment, encompasses materials such as charcoal, rice husk charcoal, and straw charcoal. Biochar has a large specific surface area and porosity, and has strong adsorption and antioxidant capacity. Application of biochar can return a large amount of nutrients to the soil and improve soil fertility [11]. Biochar has been shown to convert organic carbon fixed by plant photosynthesis into inert carbon, so that it is not rapidly mineralized by microorganisms, thereby achieving carbon sequestration and emission reduction [12]. The application of biochar has been shown to significantly alter the physical and chemical properties of soil, subsequently impacting the diversity of microbial

communities and, by extension, the mineralization of organic carbon [13]. However, the effects of biochar application on the stability of organic carbon are variable and contingent upon the biochar's characteristics, soil type, and the biotic and abiotic environment [14]. Maestrini et al. [15] discovered that fresh biochar can accelerate the mineralization of native organic carbon within the initial 20 days post-application, but may have a detrimental effect after 200 days. Wang et al. [16] found that the addition of biochar improved the stability of soil organic carbon within 0.5 years, although this effect disappeared over time. In summary, the effects of biochar and straw on soil organic carbon content and mineralization vary greatly depending on the time scale.

Soil microorganisms, acting as both drivers and regulators, exert a profound influence on the dynamics of the soil carbon cycle, with microbial activity governing the decomposition and sequestration of soil organic carbon. Furthermore, numerous bacterial communities exhibit a strong correlation with soil chemical properties, serving as reliable indicators of soil status. Investigating the relationship between the soil microbial community and soil chemical properties is instrumental in elucidating the mechanisms underlying the soil carbon cycle. Consequently, it is imperative to focus on the regulation of the soil carbon cycle and the under-lying mechanisms influenced by various fertilization practices. Research indicates that the application of straw and biochar markedly influences genes associated with soil carbon fixation and degradation [17]. Additionally, combined straw return and other fertilizer significantly enhanced soil fertility and beneficial bacterial abundance, with soil organic carbon being the primary factor influencing bacterial community structure [18]. Both amendments modulate soil $CO_2$ emissions by altering the abundance of carbon-cycle-related genes. In recent years, metagenomic techniques have advanced significantly. These techniques can be applied to obtain the genetic information of microbial communities based on the DNA of all microor-ganisms in environmental samples by non-culture-based means [19]. Research has demon-strated that fertilization enhances the prevalence of amylase genes, which are integral to starch hydrolysis and carbon decomposition pathways [20]. Zhang et al. [21] reported that the appli-cation of corn straw biochar significantly altered the structure of microbial communities asso-ciated with carbon fixation genes, enhancing the stability of soil organic carbon and suppressing the activity of carbon degrading microorganisms. Li et al. [17] observed that the application of straw had no discernible impact on genes involved in the soil carbon cycle. Li et al. [22] noted a significant upregulation of the soil carbon sequestration gene (cbbL) in response to straw or biochar amendments. In recent years, differential impacts of straw and biochar applications on the soil carbon pool have been observed, yet the mechanisms by which these amendments alter the distribution of soil organic carbon mineralization and the compo-sition of carbon-cycle-related microbial communities and genes in agricultural soils remain to be elucidated.

This study presents the findings of a five-year field experiment investigating the impact of straw and biochar on the composition of soil organic carbon and the carbon cycle within yel-low paddy soils of Guizhou Province, China. We analyzed the influence of straw and biochar on the abundance of pivotal genes involved in soil organic carbon mineralization and the car-bon cycle using a combination of indoor cultivation and metagenomic sequencing, thereby elucidating the interplay between soil organic carbon and carbon functional genes. This study is designed to evaluate the long-term effects of straw and biochar amendments on soil carbon cycling, with the objective of identifying fertilization strategies that can enhance soil carbon sequestration and reduce carbon emissions. Our aim is to determine the optimal agricultural management practices that not only improve soil quality and boost agricultural productivity but also contribute to the challenges of global climate change mitigation. "We posited the fol-lowing hypotheses:" (1) As exogenous carbon sources, both straw and biochar will enhance the

levels of soil organic carbon and its active fractions. (2) Biochar, being relatively inert, will curtail the mineralization of soil organic carbon, whereas straw, introducing more active organic carbon, will expedite the mineralization process. (3) Straw will primarily foster the degradation of organic carbon by elevating the prevalence of degradation-associated genes, whereas biochar will mitigate this degradation by enriching genes within the carbon fixation pathways.

## Materials and methods

### Soil and experimental design

The study was conducted at an experimental site situated in Tangtou Town, Sinan County, Tongren City, Guizhou Province (coordinates 108°11'35"E, 27°45'35"N). The site is located within the humid subtropical monsoon climate zone at an altitude of 399.0 m, with an average annual temperature of 17.5°C and an average annual precipitation of 1200 mm. The experimental field was managed for single-season rice cultivation on paddy soil. The experiment commenced in 2019 for a five-year period, with the same quantities of straw, biochar, and chemical fertilizer being applied each year. The initial chemical properties of the 0–20 cm soil layer were as follows: pH 6.16, organic matter 23.65 g/kg, total nitrogen 1.39 g/kg, alkali-hydrolyzed nitrogen 133.00 mg/kg, available phosphorus 37.16 mg/kg, and available potassium 182.07 mg/kg.

The experimental design included four treatments, each with three replicates: (1) control (CK) with no fertilizer application; (2) chemical fertilizer (NPK) applied at rates of N 150 kg/hm$^2$, $P_2O_5$ 148 kg/hm$^2$, and $K_2O$ 230 kg/hm$^2$; (3) straw (NPKS) integrated with chemical fertilizer, with 15 t/hm$^2$ returned to the field; and (4) biochar (NPKB) integrated with chemical fertilizer, with 4 t/hm$^2$ returned, derived from the carbonization of 15 t of straw (Table 1). The experiment was laid out in 12 plots, each measuring 30 m$^2$, arranged in a randomized block design. Phosphorus fertilizer and biochar were applied as a single basal dressing; nitrogen was split with a 5:5 basal to topdressing ratio, further divided into a 2:3 ratio for tillering and panicle fertilization; potassium followed a 5:5 basal to topdressing ratio, with topdressing occurring at the tillering stage. Field management practices were aligned with local agricultural norms. The tested fertilizers were urea containing 46.2% N, calcium superphosphate containing 16% $P_2O_5$, and potassium chloride containing 60% $K_2O$. The biochar, derived from rice straw, was manufactured by Jiangsu Province, China Nanjing Qinfeng Zhongcheng Biomass New Material Co., Ltd. Key characteristics of the biochar were a carbonization temperature of 450°C, pH 8.65, an organic carbon content of 344.97 g/kg, total nitrogen 5.99 g/kg, total phosphorus 1.99 g/kg, and total potassium 27.15 g/kg. The rice straw contained an organic carbon content of 309.72 g/kg, total nitrogen 5.99 g/kg, total phosphorus 1.87 g/kg, and total potassium 25.89 g/kg.

Soil sampling was conducted in September 2023, coinciding with the rice maturation phase. Soil samples from 0–20 cm depth were collected using the standard five-point sampling technique [23]. Samples were sieved to remove visible roots and plant residues, then homogenized and portioned into three distinct subsets: one fresh soil sample was sifted through a

**Table 1. Fertilization amount of different treatments//kg•hm$^{-2}$.**

| Treatment | Nitrogen (N) (kg/hm$^2$) | Phosphorus ($P_2O_5$) (kg/hm$^2$) | Potassium ($K_2O$) (kg/hm$^2$) | Straw (t/hm$^2$) | Biochar (t/hm$^2$) |
|---|---|---|---|---|---|
| CK | 0 | 0 | 0 | 0 | 0 |
| NPK | 150 | 148 | 230 | 0 | 0 |
| NPKS | 150 | 148 | 230 | 15 | 0 |
| NPKB | 150 | 148 | 230 | 0 | 4 |

2-mm sieve and stored in a refrigerator at 4°C for soil organic carbon mineralization test, soil soluble organic carbon and microbial biomass carbon determination; another was sifted through a 2-mm sieve and stored in a refrigerator at −80°C for metagenomic sequencing. After air-drying, the third subset was sieved through a 0.149-mm mesh for the determination of soil organic carbon and total nitrogen, and a 1-mm mesh for assessing soil pH, available nitrogen, phosphorus, and potassium.

## Experimental site and permit requirements

The experiments for this study were conducted in Tangtou Town, Sinan County, Tongren City, Guizhou Province (coordinates 108°11'35"E, 27°45'35"N), which is located in a humid subtropical monsoon climate zone. According to the relevant laws and regulations of China, the experimental site is a general agricultural area and does not involve nature reserves, eco-logical red-line areas, or other areas that require special permits. Therefore, no special field access permits were required for this experiment. During the experiment, we followed local agricultural management practices and coordinated with local agricultural departments to ensure the smooth progress of the experiment. All experimental operations complied with local environmental protection and agricultural practice standards and did not cause adverse effects on the environment.

## Determination of active components of soil organic carbon

Soil nutrient determinations included pH, measured using the composite electrode method with a water-to-soil ratio of (1:2.5); soil organic matter and organic carbon, assessed via the $K_2Cr_2O_7–H_2SO_4$ external heating method; total nitrogen, quantified using the semi-micro Kjeldahl method; alkali-hydrolyzable nitrogen, determined using the alkali diffusion method; available phosphorus, ascertained through the $NaHCO_3$ extraction-molybdenum antimony colorimetric method [24]; and available potassium, measured using the $NH_4OAc$ extraction-flame spectrophotometry method [24]. Dissolved organic carbon and microbial biomass car-bon were quantified using the chloroform fumigation-0.5 mol/L$K_2SO_4$ extraction method [25]. Readily oxidizable organic carbon was evaluated using the 0.333 mol/L$KMnO_4$ oxidation method [26].

## Indoor constant temperature mineralization culture test

Applying the lye absorption method [27], the experimental setup included three replicates per field plot, with six control blanks incorporated, giving a total of 42 mineralization culture sys-tems established. Within each system, 30.0 g of 2 mm sieved fresh soil, pre-chilled to 4°C, was weighed into a 50 mL wide-mouth bottle, with soil moisture adjusted to 35% using distilled water. The soil was then placed in a 1000 mL culture bottle, sealed, and subjected to a pre-culti-vation phase at 25°C and 45% relative humidity in the dark for 7 days. After pre-incubation, a NaOH absorption cup containing 10 mL of 1 mol/L solution was positioned at the base of the culture flask. On the 1st, 3rd, 6th, 9th, 12th, 18th, 24th, and 30th days of culture, the soil mois-ture content was adjusted and the lye absorption cup was replaced. At this time, 2 mL of 1 mol/L $BaCl_2$ solution and 2 to 3 drops of phenolphthalein indicator were added to the lye absorption cup. Finally, 0.1 mol/L HCl solution (calibrated with anhydrous $Na_2CO_3$ solution) was used to titrate until the purple color disappeared. The amount of mineralized soil organic carbon, the rate of soil organic carbon mineralization, and the cumulative mineralization rate

of soil organic carbon were calculated by using the following equations.

$$C_n = C_{HCL}*(V_0 - V_1)*22/0.03 \qquad (1)$$

$$C_v = C_n/t \qquad (2)$$

$$C_m = C_t/S_{SOC} \qquad (3)$$

where $Cn$ represents the amount of soil organic carbon mineralized (mg/kg); $C_{HCl}$ is the concentration of hydrochloric acid (mol/L); $V_0$ is the volume (mL) of the blank titration; $V_1$ is the volume (mL) of hydrochloric acid consumed; $C_v$ is the soil organic carbon mineralization rate (mg/(kg·d)); t is the incubation time; $C_m$ is the cumulative mineralization rate of soil organic carbon (%); $C_t$ is the cumulative mineralization of organic carbon in the sample (g/kg); and $S_{SOC}$ is the soil organic carbon content (g/kg). The data obtained for C mineralization under laboratory incubation of soil for each of the four treatments were modeled against a first-order kinetic equation as follows:

$$C_t = C_0\left(1 - e^{-kt}\right) \qquad (4)$$

where $C_t$ denotes the cumulative C mineralization after t days (g/kg); $C_0$ represents the potential mineralizable pool of C (mg/kg); t is the incubation time (d); and k is the rate constant of C mineralization.

## Metagenome sequencing and gene catalogue construction

The sequencing process was carried out on the Illumina NovaSeq platform at the Majorbio Bio-Pharm Technology Co., Ltd. (Shanghai, China), following the standard protocol provided by Illumina, Inc., USA (http://www.illumina.com/). After the raw sequences had been obtained, they were subjected to demultiplexing, quality trimming, and decontamination. The optimized sequences were then assembled using the software MEGAHIT, version 1.1.2 (https://github.com/voutcn/megahit). Contigs with a length of 300 base pairs or more were selected as the final assembly results. Open reading frames within the assembled contigs were predicted using Prodigal or MetaGene (http://metagene.cb.k.u-tokyo.ac.jp/). A non-redundant gene catalog was constructed using CD-HIT, version 4.6.1 (http://www.bioinformatics.org/cd-hit/), with parameters set for 90% sequence identity and 90% coverage. The longest gene from each cluster was selected as a representative sequence. High-quality reads from each sample were aligned to the non-redundant gene catalog using SOAPaligner, version 2.21 (http://soap.genomics.org.cn/), with 95% identity to calculate the gene abundance in the respective samples.

For taxonomic annotation, the amino acid sequences of the non-redundant gene catalog were aligned against the NR database using Diamond, version 0.8.35 (http://www.diamondsearch.org/index.php), with an e-value cutoff of 1e−5. The species annotations were obtained from the corresponding taxonomic information in the NR database, and the total gene abundance for each species was calculated. Functional annotations were performed by aligning the amino acid sequences of the non-redundant gene catalog against the KEGG database (https://www.genome.jp/kegg/) using the same version of Diamond, with the same e-value cutoff. Genes corresponding to KEGG functions were identified, and the total gene abundance for the KO, Pathway, EC, and Module categories was calculated to determine the abundance of the corresponding functional categories.

## Statistics and data analysis

Data were statistically summarized using Excel, variance analysis, Duncan's multiple range test, first-order kinetic modeling, and correlation heatmaps and histograms were generated using Origin 9.0 software. Metagenomic sequencing data analysis was performed with the I-Sanger Bioinformatics Cloud Platform, available at I-Sanger Bioinformatics Cloud Platform (https://www.majorbio.com/). Species community, diversity indices, and analysis charts depicting the contributions of species and genes were generated. Partial least squares path modeling (PLS-PM) was conducted using the 'plspm' package within R version 3.6.1. Correlation analysis between genes and species was graphically represented using Gephi 1.0 software.

## Results

### Effects of return of straw and carbonized material to the field on soil organic carbon and its components

The soil organic carbon (SOC) levels were significantly elevated by 73% and 36% under the NPKS and NPKB treatments, respectively, compared with the CK treatment ($p < 0.05$, for all subsequent comparisons) (Table 2). The addition of straw and biochar to the soil led to a substantial increase in dissolved organic carbon (DOC) levels, with the NPKS and NPKB treatments raising DOC by 38% and 23%, respectively, compared to the control, yet no significant difference was observed between the two treatments. In the NPKS and NPKB treatments, the microbial biomass carbon (MBC) was significantly increased by 40% and 20% compared to the control, respectively. While readily oxidizable carbon (ROC) levels remained relatively consistent across all treatments, the NPKS treatment recorded the highest values, with the biochar-amended and chemical treatments following suit. The composition of active SOC components was affected by the fertilization methods, with the NPKS treatment exerting the most notable impact. This particular treatment led to a significant reduction in the DOC/SOC ratio by 19% when compared to the control, a change not seen with other treatments. Despite no significant variations in the MBC/SOC ratio among the treatments, the straw-enriched treatment had the lowest MBC/SOC ratio, with the biochar-amended and chemical treatments showing reductions of 18% and 12%, respectively, compared to the control. Moreover, the straw-enriched treatment caused a significant 35% decrease in the ROC/SOC ratio in comparison to the control. These findings suggest that specific fertilization strategies can alter the dynamics of SOC components, potentially influencing soil fertility and carbon sequestration potential.

### Effects of return of straw and carbonized material to the field on soil organic carbon mineralization

**Mineralization rate and cumulative amount of soil organic carbon.** Under the various fertilization treatments, the rate of soil organic carbon mineralization typically exhibited an

**Table 2. Soil carbon components under different fertilization treatments.**

| Treatments | SOC (g/kg) | DOC(mg/kg) | MBC(mg/kg) | ROC(g/kg) | DOC/SOC (%) | MBC/SOC (%) | ROC/SOC (%) |
|---|---|---|---|---|---|---|---|
| CK | 13.75±0.28c | 159.9±4.35b | 190.43±13.94c | 4.82±0.53a | 1.16±0.02a | 1.38±0.11a | 35.04±3.14a |
| NPK | 15.05±0.55c | 176.3±5.79b | 189.02±3.88c | 4.93±0.10a | 1.17±0.05a | 1.25±0.02a | 32.77±1.76ab |
| NPKS | 23.75±0.62a | 221.4±12.30a | 266.28±19.08a | 5.36±0.32a | 0.93±0.04b | 1.12±0.09a | 22.58±0.79c |
| NPKB | 18.71±1.00b | 196.8±7.26a | 228.00±6.04ab | 5.16±0.79a | 1.05±0.11a | 1.21±0.08a | 27.55±4.56bc |

Note: Data are means ± standard errors, and different lowercase letters indicate significant differences among treatments ($p < 0.05$). SOC, soil organic carbon; DOC, dissolved organic carbon; MBC, microbial biomass carbon; ROC, readily oxidizable carbon

initial decline followed by stabilization over time (Fig 1b; Table 3), fitting well with the logarithmic function y = a + b ln(x), ($p < 0.01$). The mineralization process of soil organic carbon can be divided into two phases: an initial rapid decline and subsequent stabilization. The initial 9 days marked the decline phase, with the $CO_2$ release rate peaking within the first 1 to 3 days and subsequently experiencing a rapid decrease. The $CO_2$ release rate tended to be stable from day 9 to day 30. Compared with day 1 levels, the mineralization rates decreased by 35%–58% and 45%–69% by days 9 and 30, respectively. Throughout the incubation, the NPKS treatment exhibited the highest mineralization rate of 187.00–51.94 mg/(kg·d), with the NPK, NPKB, and CK treatments displaying rates of 139.33–32.96 mg/(kg·d), 70.89–40.94 mg/(kg·d), and 88.00–32.71 mg/(kg·d), respectively. The cumulative mineralization of soil organic carbon increased through time under all fertilization treatments, with the release rate transitioning from rapid to gradual (Fig 1c). The NPKS treatment displayed the highest cumulative $CO_2$ release of 2.35 g/kg, followed by the NPK treatment with 1.74 g/kg, the NPKB treatment with 1.71 g/kg, and the CK treatment with 1.47 g/kg. After the end of incubation, the NPKS, NPK, and NPKB treatments had increased the cumulative mineralization of soil organic carbon by 59%, 21%, and 16%, respectively, relative to the CK treatment.

The cumulative mineralization rate of soil organic carbon can effectively reflect the carbon sequestration capacity of soil: the lower the ratio, the stronger the carbon sequestration capacity of soil, and vice versa. At the end of the culture period, the difference in the cumulative mineralization rate of soil organic carbon between different fertilization treatments was obvious (Fig 1a). The NPKB treatment exhibited the lowest ratio, and the NPK treatment the highest ratio. Relative to the CK treatment, the ratios of the NPKB and NPKS treatments were lower by 14% and 7%, respectively. The ratio for the NPK treatment was greater than that of the CK treatment by 8%, which weakened the carbon sequestration capacity of soil.

**Dynamic equation parameters and regression equation of soil organic carbon mineralization.** A first-order kinetic equation was used to fit the relationship between the cumulative mineralization of organic carbon and the incubation days for the different fertilization treatments as follows: $C_t = C_0 (1 - e^{-kt})$ (Table 2). Here, $C_0$ denotes the potential mineralizable organic carbon in soil. The NPKS and NPKB treatments increased the amount of soil potential organic carbon mineralization compared with the CK treatment, by 38% and 27%, respectively. The NPK treatment decreased soil potential organic carbon mineralization by 8%

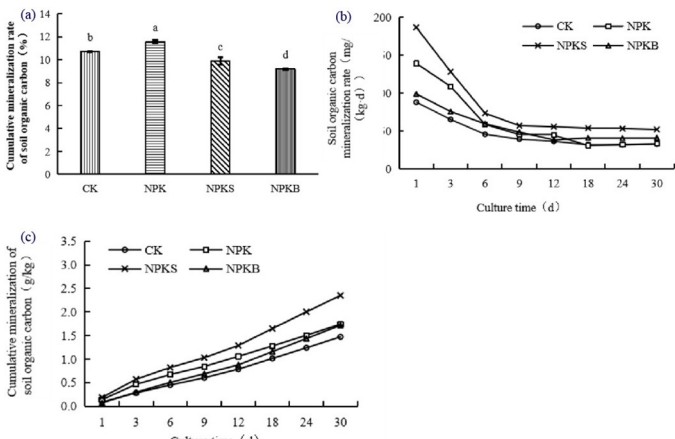

**Fig 1. Effects of different fertilization treatments on soil organic carbon mineralization.** (a) Soil organic carbon mineralization rate; (b) Cumulative mineralization of soil organic carbon; (c) Cumulative mineralization rate of soil organic carbon.

**Table 3. Soil organic carbon mineralization kinetic equation and its parameters.**

| Treatments | $C_t$(g/kg) | $C_0$ (g/kg) | k (d$^{-1}$) | $T_{1/2}$ (d) | $C_0$/SOC (%) | $R^2$ | Regression equation | $R^2$ |
|---|---|---|---|---|---|---|---|---|
| CK | 1.47 | 2.11 | 0.038 | 18.14 | 15.3 | 0.995** | y = 83.205−7.83ln(x) | 0.939** |
| NPK | 1.74 | 1.93 | 0.066 | 10.50 | 12.8 | 0.984** | y = 135.16−5.57ln(x) | 0.930** |
| NPKS | 2.35 | 2.92 | 0.050 | 13.86 | 2.2 | 0.980** | y = 170.26−6.19ln(x) | 0.887** |
| NPKB | 1.72 | 2.70 | 0.032 | 21.65 | 4.4 | 0.996** | y = 95.041−9.95ln(x) | 0.938** |

Note: (1) $C_t$ is cumulative mineralization of organic carbon; (2) $C_0$ is the amount of potentially mineralizable organic carbon; (3) k is the rate constant of organic carbon mineralization; (4) $T_{1/2}$ is the half turnover period; $C_0$/SOC is the ratio of potential mineralized organic carbon to total organic carbon in soil.

relative to the CK treatment. The turnover rate (k) and half turnover period ($T_{1/2}$) of the soil organic carbon pool were also different under different fertilization treatments. The k value was between 0.032 and 0.066 d$^{-1}$, and the $T_{1/2}$ value was between 10.5 and 21.6 d (Table 2). The turnover rate of the NPK treatment was the fastest, followed by the NPKS, CK, and NPKB treatments, and the half turnover period exhibited the opposite pattern. The straw (NPKS) and biochar (NPKB) treatments were found to promote soil potential organic carbon mineralization relative to the CK treatment, with the NPKS treatment showing the most pronounced enhancement; in contrast, the NPK treatment did not favor increased potential mineralizable organic carbon.

### Effects of return of straw and carbonized material to the field on soil carbon cycling genes and microbial communities

Different fertilization treatments exerted varying influences on the abundance of carbon degradation genes. We assessed the abundances of functional genes involved in starch degradation, hemicellulose degradation, cellulose degradation and chitin degradation (Fig 2). Among the starch degradation genes, the *ISA* gene exhibited the highest abundance and a statistically significant difference among treatments ($p < 0.05$). The relative abundance of starch degradation genes varied among fertilization treatments: Compared to the control group (CK treatment), the NPKS treatment increased the relative abundance of the *ISA* and *malQ* genes by 7.3% and 0.4%, respectively. Similarly, the NPKB treatment also enhanced the relative abundance of the *malZ* gene by 6.3% compared to the control group. As carbon degradation progressed, the *rfbB* gene, associated with hemicellulose degradation, displayed the highest abundance, with the CK treatment showing the most pronounced effect with a relative abundance of 50.64%. Furthermore, fertilization treatments led to an increase in the *xynC* gene abundance from 0% to 2.4%. The NPKS treatment yielded the highest abundance of *bglX* and *celF* genes related to cellulose degradation with relative gene abundances of 58.63% and 3.84%,

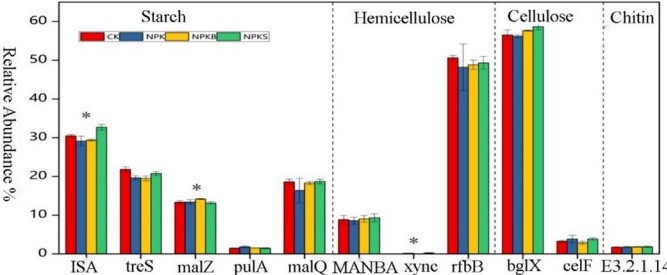

**Fig 2. Relative abundances of carbon degrading genes under different fertilization treatments.**

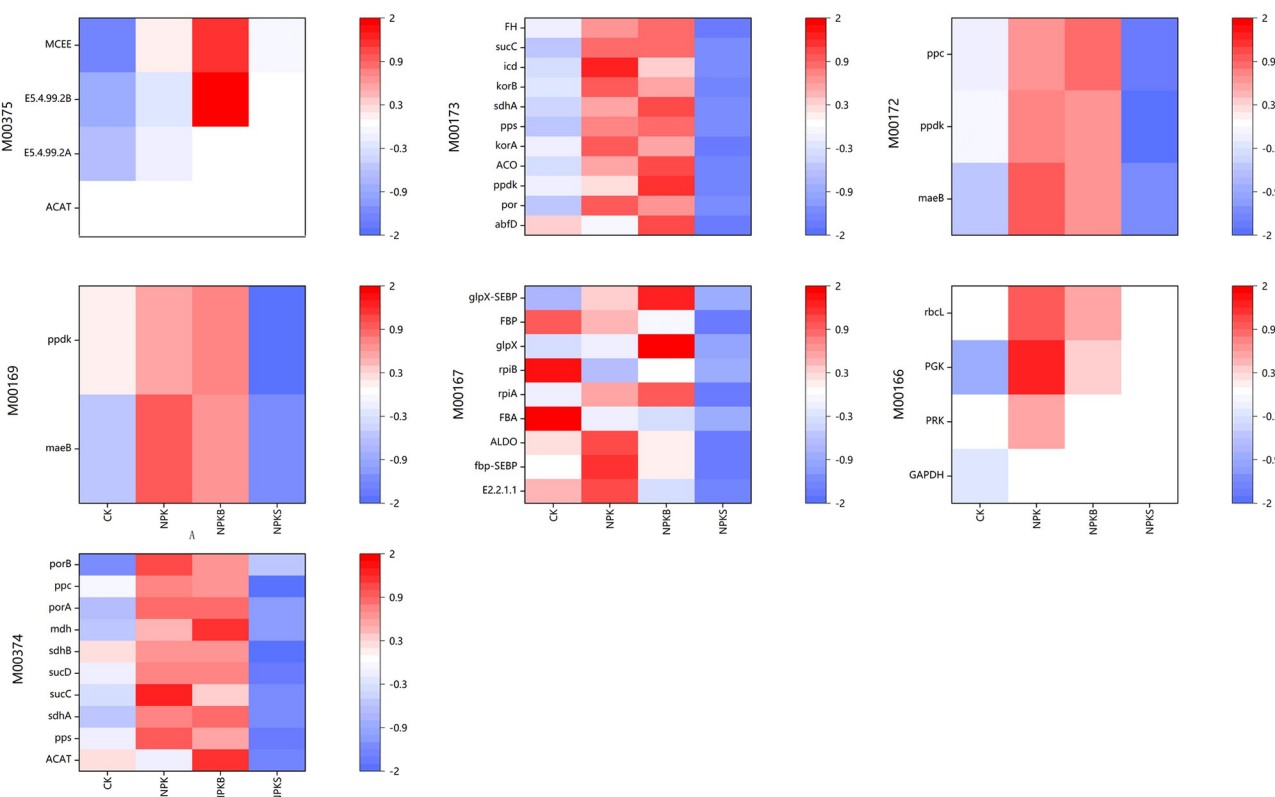

**Fig 3. Relative abundances of carbon fixation genes under different fertilization treatments.** Calvin cycle (M00166); ribulose-5p to glyceraldehyde-3p (M00167); CAM cycle (M00169); c4-dicarboxylic acid cycle(M00172); rTCA cycle (M00173); DC/4-HB cycle (M00374); and 3-HP/4-HB cycle (M00375). The color gradient from blue to white to red in the figure denotes the increasing relative abundance of genes, ranging from low to high.

respectively; however, no significant differences were observed. Among the chitin degradation genes, endochitinase (E3.2.1.14) demonstrated the highest abundance under the NPKS treatment, reaching 1.84%, compared to 1.74%, 1.76%, and 1.81% in the control (CK), NPK, and NPKB treatments, respectively. In conclusion, the impact of various fertilization treatments on the degradation of labile carbon is complex. Fertilization treatments can expedite the breakdown of active organic carbon, whereas straw amendment appears to promote the degradation of recalcitrant carbon.

A total of 101 carbon fixation-associated functional genes were identified across various fertilization treatments. The genes were categorized based on carbon fixation pathways, including the Calvin cycle (M00166), the ribulose-5-phosphate to glyceraldehyde-3-phosphate pathway (M00167), the aspartate metabolic cycle (CAM; M00169), the C4-dicarboxylic acid cycle (M00172), the rTCA cycle (M00173), the DC/4-HB cycle (M00374), and the 3-HP/4-HB cycle (M00375) Fig 3 Relative abundances of carbon fixation genes under different fertilization treatments: Calvin cycle (M00166); ribulose-5p to glyceraldehyde-3p (M00167); CAM cycle (M00169); c4-dicarboxylic acid cycle). In the rTCA cycle (M00173), compared to the CK treatment, the NPKB and NPK treatments increased the abundance of 11 associated genes, including *ppdK*, *ACO* and *por*, whereas the NPKS treatment resulted in a decrease in their abundance. The NPKS treatment increased the abundances of the *MCEE* gene in the 3-HP/4-HB cycle (M00375) and of the *GAPDH* genes in the Calvin cycle (M00166), raising the relative abundance by 6.0% and 8.3%, respectively, compared to the CK treatment. Furthermore,

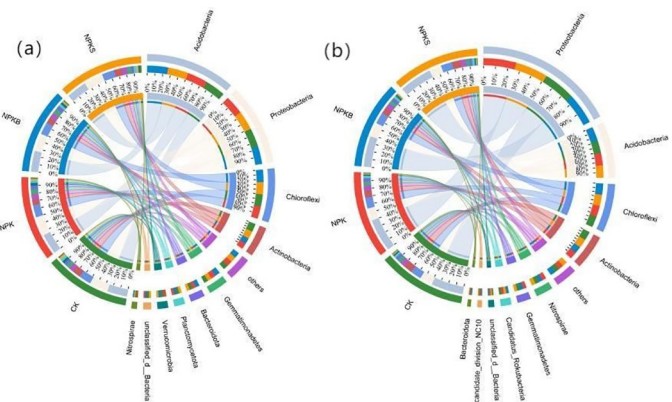

**Fig 4. Effects of different fertilization treatments on soil bacterial abundance.** (a) Abundance of Carbon Degrading Bacteria; (b) Abundance of Carbon Fixing Bacteria.

in the ribulose-5-phosphate to glyceraldehyde-3-phosphate pathway (M00167), the NPKB treatment enhanced the abundances of the *glpX*, *rpiA* and *glpX-SEBP* genes by 15%, 10% and 13%, respectively, compared to the CK treatment. Concurrently, it decreased the abundances of the *FBP* and *rpiB* genes by 9% and 7.2%, respectively. The gene abundances in other fixed pathways were generally higher in the NPKB and NPK treatments than in the CK and NPKS treatments. In conclusion, the application of straw does not favorably influence the abundance of genes related to carbon fixation, whereas biochar application effectively enhances such gene abundance and decreases carbon emissions.

The effects of different fertilization treatments on the carbon fixing and carbon degrading microbial communities are shown in Fig 4. For carbon fixation genes, the predominant phyla include Proteobacteria, Acidobacteria, Chloroflexi, Actinobacteria, Nitrospira, Gemmatimonadetes, and Bacteroidota (Fig 4a). The NPKB treatment resulted in a higher relative abundance of Acidobacteria, reaching 26%, compared to the 25% observed under the NPKS treatment. Relative to the control treatment, the NPKS treatment increased the relative abundance of Proteobacteria by 4%, while the NPKB treatment caused a 30% reduction in the relative abundance of Actinobacteria. For the carbon degradation genes, the dominant phyla were Acidobacteria, Proteobacteria, Chloroflexi, Actinobacteria, Gemmatimonadetes, Bacteroidota, Planctomycetota, Verrucomicrobia, and Nitrospirae (Fig 4b). Compared to the CK treatment, the NPKB treatment resulted in a 4% reduction in the relative abundance of Proteobacteria, amounting to 25%. The NPKB treatment also induced a 4% increase in the relative abundance of Acidobacteria, while the NPKS treatment had a minimal effect on the relative abundance of Acidobacteria. Chloroflexi showed an increase in relative abundance across different fertilization treatments, whereas the NPKB treatment decreased that of Actinobacteria compared with the CK treatment. Furthermore, the diversity index analysis presented in Fig 5 revealed distinct microbial community structures associated with carbon sequestration and degradation among the different fertilization treatments.

The proportional contributions of the foremost ten functional genes to the microbial communities participating in the carbon degradation and fixation cycles are illustrated in Fig 6. A diminution in the prevalence of microbial communities is accompanied by shifts in the assembly of the respective functional genes. Proteobacteria contribute most markedly to carbon fixation gene functions, and Acidobacteria are identified as having the greatest impact on the functionality of carbon degradation genes. The influence of Acidobacteria on the recalcitrant

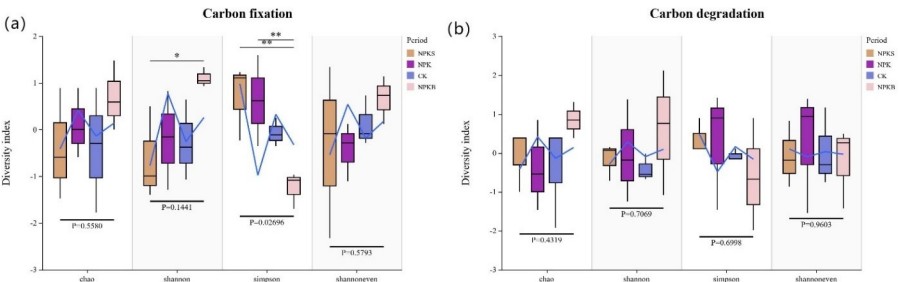

**Fig 5. Soil bacterial diversity indices in response to fertilization.** (a) Carbon Fixation Bacteria; (b) Carbon Degrading Bacteria.

carbon degradation gene, blgX, is both maximal and positively efficacious (Fig 7). The microbial communities exert the most significant influence on the M00374 pathway within the carbon fixation processes. Proteobacteria, recognized as the principal catalysts for carbon fixation, display a positive correlation with the pathways of carbon sequestration. The correlation coefficients indicate that the *ISA*, *malQ*, *MANBA*, *xynC*, *bglX*, and *E3.2.1.14* genes exhibit significant positive correlations with soil organic carbon mineralization ($p < 0.05$; Fig 8a). Notably, the *E3.2.1.14* and *xynC* genes had the most substantial influence on organic carbon mineralization, with a correlation coefficient of 0.87 in both cases. Additionally, the ISA, *malQ*, *MANBA*, *bglX*, *xynC*, *celF*, and *E3.2.1.14* genes showed significant positive correlations with soil organic carbon components, including SOC, DOC, MBC, and ROC. Among these, the *E3.2.1.14* gene had the most pronounced effect on ROC and SOC, exhibiting correlation coefficients of 0.96 and 0.91, respectively. The *MANBA* gene significantly influenced MBC, demonstrating a correlation coefficient of 0.93(Fig 8a). The carbon sequestration pathway exhibited significant negative correlations with organic carbon components and mineralization. PLS-PM analysis (Fig 8b)

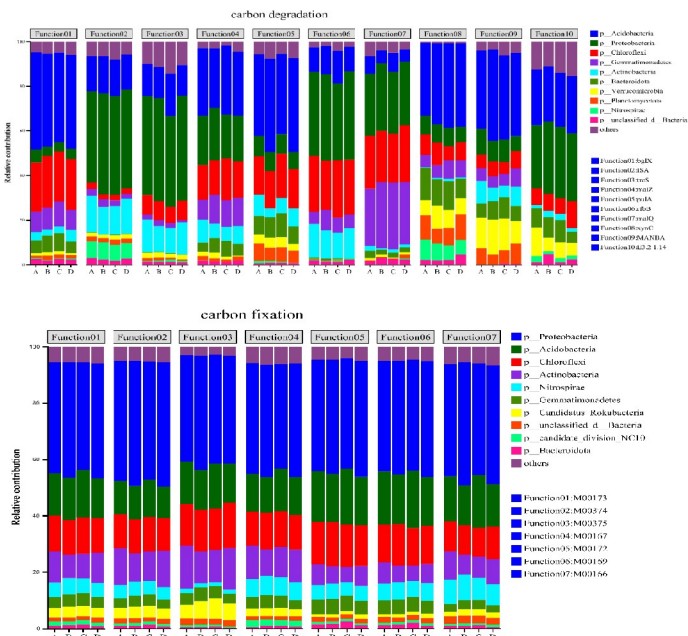

**Fig 6. Contribution analysis of carbon cycle genes and soil bacteria.** A: CK; B: NPK; C: NPKB; D: NPKS.

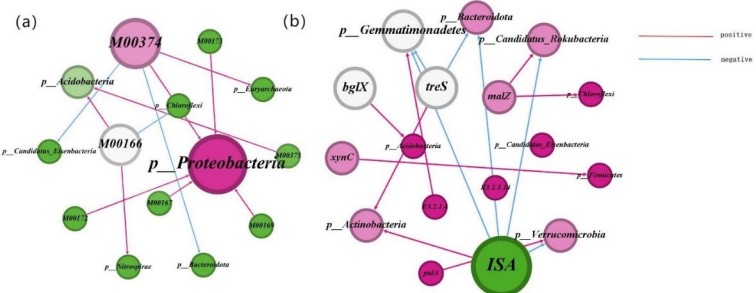

**Fig 7. Correlation analysis between microorganisms and their functional genes.** (a) Carbon Fixing Genes and Their Microorganisms; (b) Carbon Degrading Genes and Their Microorganisms.

demonstrated that the abundance of soil organic carbon fractions and functional genes related to carbon degradation were positively correlated with both the rate of organic carbon mineralization and the abundance of functional genes related to carbon fixation; in addition, microbial abundance was negatively correlated with the rate of organic carbon mineralization. A significant positive correlation was observed between the soil organic carbon fractions and the abundance of functional genes within the carbon cycle.

## Discussion

### Effects of return of straw and carbonized material to the field on soil organic carbon components

Soil organic carbon plays a pivotal role in the Earth's carbon cycle and soil fertility, and constitutes approximately 80% of the total carbon within the terrestrial carbon cycle [28]. The concentration of organic carbon in agricultural soils is influenced by various factors, including fertilization practices, climatic conditions, land use types, and vegetation cover [29]. This study found that the NPKS and NPKB treatments can effectively improve the soil organic

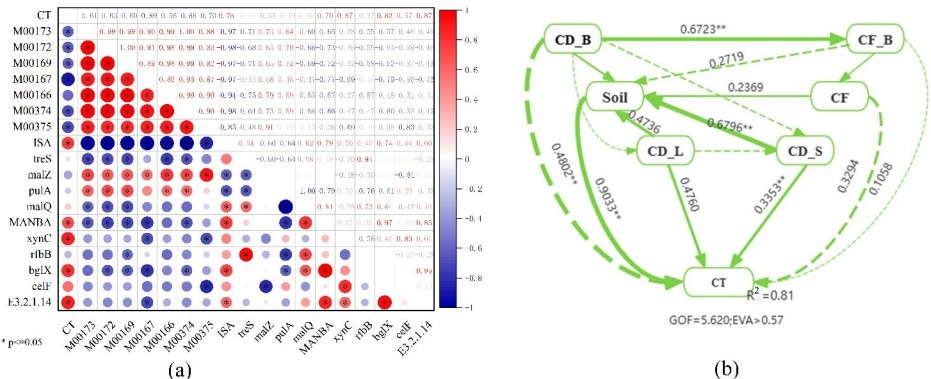

**Fig 8. Analysis of drivers for soil organic carbon mineralization.** (a) Correlation Analysis of Carbon Sequestration Pathways and Degradation Genes with Soil Carbon Mineralization and Active Carbon Components; (b) Partial Least Squares Path Modeling: CT (Accumulated Mineralization of Soil Organic Carbon); cd_L (Abundance of Functional Genes for Labile Carbon Components such as Starch and Hemicellulose); cd_S (Abundance of Functional Genes for Stable Carbon Components such as Cellulose and Chitin); CF_S (Abundance of Carbon Sequestration Functional Genes); Soil (Soil Organic Carbon Components); CD_B (Relative Abundance of Carbon Fixing Bacteria); CF_B (Relative Abundance of Carbon Fixing Bacteria); Solid Lines Indicate Positive Correlations, Dashed Lines Indicate Negative Correlations.

carbon content (Table 2). This finding is consistent with the study of Zou et al [30]. On the one hand, the incorporation of straw as an exogenous carbon source into the soil can induce the priming effect, thereby increasing the content of native soil organic carbon; on the other hand, return of straw can enhance the activity of soil microorganisms, enhance mineralization of organic carbon, and promote the increased organic carbon storage while adding a large amount of carbon into the soil [31]. Furthermore, the combination of straw with chemical fertilizers can facilitate the formation of aggregates and organic–inorganic complexes, thereby reducing microbial decomposition of organic carbon and enhancing the physical protection of soil aggregates for organic carbon. Application of biochar produced by straw carbonization can effectively raise the organic carbon content in the soil and increase the accumulation of the soil carbon pool [32]. Biochar, which is characterized by a high carbon content, can directly augment the soil carbon content upon application. The material possesses the capacity to suppress the decomposition of organic carbon, moderating the decomposition rate through the provision of a stable carbon source and by ameliorating the soil microenvironment, thus extending the residence time of soil organic carbon. In addition, the intervention of biochar can change the physical and chemical properties of soil, markedly improve the water retention capacity and aggregate stability of soil, promote interactions between soil microorganisms and organic matter, and increase the fixation and storage effect of carbon [33]. Significant variations in soil organic carbon content were observed following the application of straw and biochar, potentially attributable to differences in the turnover rates of the respective organic materials (Table 3). After 5 to 10 years of biodegradation, straw retains approximately 10%–20% of the initial carbon input within the field, whereas carbonized straw retains only ca. 3% of this input [27]. "Soil active organic carbon" typically refers to the organic carbon fractions in the soil that are relatively easily decomposable and participate in the soil carbon cycle. These fractions include, but are not limited to, Dissolved Organic Carbon (DOC), Microbial Biomass Carbon (MBC), and Readily Oxidizable Carbon (ROC) [23]. Although soil active organic carbon constitutes a minor fraction of the total organic carbon, it is highly responsive to soil management practices and serves as a sensitive bioindicator of soil quality and nutrient cycling dynamics [23]. In this study, the NPKS treatment exhibited the highest levels of DOC, MBC, and ROC, followed by the NPKB and NPK treatments (Table 2), corroborating the results of Wang [34] and Yang [27]. However, while the NPKS and NPKB treatments significantly increased microbial biomass carbon (MBC), we observed no significant difference in the rate of soil organic carbon turnover (k value) between the NPKB treatment and the control (Table 3). This unexpected outcome may be attributed to several factors. Firstly, under the NPKS treatment, the decomposition of straw rapidly releases nutrients that can be quickly utilized by microorganisms, thereby increasing microbial biomass [35]. In contrast, biochar in the NPKB treatment, due to its high specific surface area and porosity, effectively adsorbs and retains organic carbon and nutrients in the soil, reducing their loss. Moreover, the high chemical stability of biochar makes it less susceptible to microbial decomposition, which may slow the turnover rate of soil organic carbon, resulting in no significant increase in the carbon turnover rate (k value) even with high MBC [36]. Additionally, the incorporation of biochar may alter the structure of the soil microbial community, potentially promoting the growth of microorganisms with slower decomposition rates, thus lowering the overall mineralization rate. These findings indicate that the impact of different organic materials on soil carbon cycling is complex and influenced by multiple factors [37]. This finding suggests that integrating chemical fertilizer with application of straw can markedly enhance the levels of soil active organic carbon components, outperforming the combination of chemical fertilizer with biochar. The application of straw likely stimulates the activity of crop roots and soil microorganisms, enhancing metabolite secretion and consequently elevating the content of soil active

organic carbon. The chemical structure of biochar, characterized by high stability and porosity, inhibits further microbial decomposition and slows the organic matter degradation rate, thereby extending the decomposition process and enhancing soil carbon stability [38]; as a result, the quantity of active organic carbon may be lower compared with that observed with the direct application of straw. We observed that the ratio of DOC, MBC, and ROC to total organic carbon was lowest in the NPKS treatment, a finding that contrasts with the outcomes reported by Zhao [39] and Xu [40]. This decrease may be attributed to a disproportionate increase in soil organic carbon content following straw application, with the rate of increase for other active components lagging behind that of the total organic carbon quantity. Furthermore, biochar's inherent adsorptive properties may reduce the concentration of active carbon components in the soil, consequently lowering the ratio.

## Effects of return of straw and carbonized material to the field on soil organic carbon mineralization

The mineralization of soil organic carbon to release $CO_2$ under the action of microorganisms is an important part of the carbon cycle. This study revealed a logarithmic relationship between the mineralization rate of soil organic carbon and time, modeled by $y = b + k \cdot \ln(t)$ ($p < 0.01$) (Table 3), indicating that a 1% change in the incubation period corresponds to an absolute change of k percent in the mineralization rate of organic carbon. Furthermore, the mineralization rate of soil organic carbon across treatments exhibited an initial downward trend over time, eventually stabilizing, a pattern consistent with the findings of Shi et al. [41]. This decline occurs because, in the early stage of mineralization, more active organic carbon is easily decomposed, and soil microbial activity is strong, leading to a high mineralization rate. In the later stage, the carbon source available for microbial mineralization is reduced, microbial activity is weakened, organic carbon mineralization is limited, and the mineralization rate gradually decreases and tends to be stable [42]. In this study, the application of straw resulted in a significantly higher mineralization rate compared with biochar and the control treatment. The application of straw may have increased the carbon and nitrogen sources that could be directly used, such as active organic carbon, cellulose and sugar, and provided sufficient nutrient sources for microbial activities. Biochar, with its strong adsorption capacity and abundant inert organic carbon, does not supply readily available carbon sources for microorganisms but instead mitigates the degradation of native soil organic carbon, effectively immobilizing soil organic carbon over extended periods [27]. The cumulative mineralization of soil organic carbon over time adheres to a first-order kinetic equation, described as $C_t = C_0 (1 - e^{-kt})$ (Table 3). The results showed that straw combined with chemical fertilizer could significantly increase the potential mineralizable organic carbon ($C_0$), reduce the stability of soil organic carbon, and promote the mineralization of soil organic carbon. Biochar combined with chemical fertilizer could slow down the turnover rate of soil organic carbon (k), increase the turnover time of soil organic carbon ($T_{1/2}$), and inhibit the mineralization of soil organic carbon. This effect may stem from the direct introduction of nutrients through straw return, which enhances soil microbial activity and elevates the soil's content of potentially mineralizable organic carbon ($C_0$). In contrast, biochar introduces stable aromatic carbon that resists microbial decomposition and utilization, thereby contributing to the soil's long-term carbon sequestration [27].

## Effects of return of straw and carbonized material to the field on soil carbon cycling genes and microbial communities

Microorganisms play a dual role in the soil carbon pool dynamics: on the one hand, they release carbon dioxide into the atmosphere through decomposition and metabolic processes;

on the other hand, they transform carbon into recalcitrant forms that are resistant to decomposition [43]. Straw and biochar differentially affect soil organic carbon, leading to variations in the abundance of key functional genes. We examined the relative abundances of functional genes involved in the degradation of starch, hemicellulose, cellulose, and chitin (Fig 2). Changes in the abundances of functional genes for the decomposition of active organic carbon substrates, such as starch and hemicellulose, were complex; however, the NPKS treatment specifically increased the abundance of genes for the decomposition of recalcitrant carbon substrates, including cellulose and chitin. Furthermore, correlation analysis (Fig 8) indicated that among the starch-degrading genes, *ISA* had the most significant impact on organic carbon mineralization, followed by *XYNC* for hemicellulose, *bglX* for cellulose, and *E3.2.1.14* for chitin. Notably, the NPKS treatment exhibited the highest relative abundances of these genes, suggesting that straw application is more effective in enhancing the abundance of carbon degrading functional genes than other fertilization strategies. In terms of degradation of active organic carbon, straw and biochar as external carbon sources increased the nutrients of soil organic C, N and other microbial growth, increased soil hydrolase activity, provided more C sources for soil microorganisms, and promoted the increase of carbon degradation functional genes [44, 45]. Additionally, the material composition of straw and biochar varies. After application to the soil, the production of soil-specific degrading enzymes may differ on different substrates. Deng et al. [3] found that biochar treatment increased cellulase activity, produced glucose, and provided ready-made carbon for microbial growth. In the context of recalcitrant carbon degradation, the adsorptive and protective properties of biochar can diminish microbial growth and activity [46, 47]. Some studies indicate that soil amendment with biochar can reduce the activity of enzymes involved in the carbon cycle [43, 48], whereas straw application can expedite the degradation of recalcitrant compounds such as cellulose and chitin. This effect may be attributed to increases in the soil content of cellulose and chitin as a result of straw application, enhancing the activity of their respective degrading enzymes, and thus fostering the proliferation of related microbial populations [49].

The impact of various fertilization treatments on carbon fixation pathways, as represented by genes, exhibits inconsistencies (Fig 3). We evaluated seven carbon fixation pathways identified in paddy soil within the KEGG database. Correlation analysis (Fig 8) demonstrated a negative association between carbon fixation genes and the mineralization of organic carbon. Notably, the metabolic pathway converting ribulose-5-phosphate to glyceraldehyde-3-phosphate (M00167) exerted the most significant influence on the mineralization of soil organic carbon. In comparison to the ribulose-5-phosphate to glyceraldehyde-3-phosphate pathway (M00167), the Calvin cycle is more energetically demanding. The Calvin cycle is the main pathway for carbon dioxide fixation, which catalyzes the carboxylation of ribulose-1,5-bisphosphate into two molecules of glyceryl 3-phosphate. Previous reports indicated no impact of long-term fertilization on the Calvin cycle [50]; however, in this study we observed lower gene abundance in the NPKS treatment compared with the CK treatment for the Calvin cycle (M00166). Furthermore, gene abundance in the remaining carbon fixation pathways revealed that the NPKB and NPK treatments exhibited significantly higher levels than the CK and NPKS treatments, suggesting that straw application may attenuate carbon fixation in soil. This is because the high C:N ratio, crystallinity, and cellulose and lignin content of straw provide nutrients for microbial degradation of soil carbon, thereby reducing carbon fixation. Wang et al. [51] demonstrated that straw application significantly elevated the proportion of soil alkoxy carbon, signifying substantial carbon release during straw decomposition and promoting the proliferation of microorganisms associated with carbon degradation. Additionally, we found that straw increased the levels of active components in soil organic carbon (Table 2). In contrast to straw, biochar enhanced nutrient retention in soil, thereby promoting the survival

and growth of microorganisms. Consequently, biochar also contributed to elevating the abundance of autotrophic organisms capable of fixing carbon dioxide, which in turn increased the abundance of carbon fixation genes [52]. Jin et al. [53] reported a significant reduction in the activity of extracellular enzymes involved in carbon substrate hydrolysis upon biochar addition, explaining the enhanced carbon fixation efficiency following biochar application. In contrast, the abundance of carbon fixation genes in soils treated solely with chemical fertilizer exceeded that of the control treatment. This outcome is attributed to the long-term chemical fertilizer application causing a reduction in soil microbial diversity and alterations in the microbial community structure, consequently impairing the microorganisms' capacity to decompose soil carbon.

The functionality of soil microorganisms is predominantly substrate-dependent [54]. Straw is rich in cellulose, lignin, and other recalcitrant compounds; in contrast, biochar, serving as an external carbon source, supplies essential nutrients for microbial growth and modulates the soil microbial environment. The differences in the structures of these materials and their different effects on soil organic carbon mean that the abundance of the dominant phylum is different. In this study (Fig 4), the phyla Proteobacteria and Actinobacteria were most abundant, whereas Bacteroidetes had the lowest representation. NPKS treatment increased the relative abundance of Proteobacteria. This rise is attributed to the abundance of genes in Proteobacteria that encode enzymes for carbohydrate metabolism, endowing them with a diverse array of taxa capable of decomposing recalcitrant carbon sources such as lignin. Additionally, straw application increased the pool of degradable soil carbon and introduced numerous refractory compounds, such as lignin, phenols, and tannins, which foster the growth of Proteobacteria [55]. Furthermore, prior research has demonstrated that straw addition augments the quantity of soil macro-aggregates, enhancing soil permeability and thus supporting the proliferation and development of aerobic microorganisms [56]. Actinomycetes possess an array of genes for the degradation of labile carbon and exhibit a more favorable response to plant residues with elevated C/N ratios [57]. In this study, the NPKB treatment showed a reduced abundance of Actinobacteria compared with the CK treatment. This result could be because of biochar's inert nature, which, while increasing soil organic carbon content, also adsorbs essential substrates and nutrients required for microbial metabolism, thereby suppressing the proliferation of Actinobacteria. The low abundance of Bacteroidetes might be attributed to the unsuitable growth conditions provided by straw and biochar amendments. The abundance of Bacteroidetes exhibited significant variation across different environmental conditions. Bacteroidetes may be predominantly composed of polytrophic groups in one setting and oligotrophic groups in another, reflecting their adaptability to diverse environmental conditions [54].

In conclusion, the integrated application of chemical fertilizer and straw enhanced the soil's capacity to degrade recalcitrant carbon, thereby facilitating the decomposition of organic carbon. Conversely, biochar supplementation elevated the abundance of genes associated with carbon fixation, decelerated the decomposition rate of organic carbon, and extended the carbon turnover time, thereby enhancing carbon sequestration and emission mitigation.

## Conclusions

This study established that the impacts of straw and biochar on soil organic carbon vary significantly. As external carbon sources, both straw and biochar enhanced the soil organic carbon content and its active components, with straw demonstrating the most pronounced effect. Genetically, the refractory carbon degradation genes were the primary catalysts for soil organic carbon degradation, and the addition of straw activated the soil's refractory carbon pool. The long-term application of straw altered the stability of soil carbon sequestration and accelerated

the decomposition of soil organic carbon. Biochar did not significantly affect the soil's refractory carbon pool; however, it elevated the abundance of genes associated with carbon fixation, thereby inhibiting the decomposition of soil organic carbon. At the microbial taxonomic level, straw treatment increased the relative abundance of Proteobacteria, enhancing their capacity for carbon decomposition and accelerating the mineralization of organic carbon. In contrast, biochar suppressed the proliferation of Actinobacteria and diminished their capacity for stable carbon decomposition, consequently slowing the organic carbon degradation process.

## Supporting information

**S1 Table. Cumulative mineralization of soil organic carbon for 30 days and variance analysis.**
(XLS)

**S2 Table. Genes and species numbers in different fertilization treatments.**
(XLS)

## Acknowledgments

We thank research staffs for their contributions to this work.

## Author Contributions

**Resources:** Jie Wei, Shengmei Yang, Qinwen Zheng.

**Writing – original draft:** Fangchi Wang.

**Writing – review & editing:** Xiaoli Wang, Jianjun Duan, Sanwei Yang.

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
