## [Decision Letter · Decision Letter 0]

18 Oct 2024

PONE-D-24-40473The impact of straw and its post-pyrolysis incorporation on functional microbes and mineralization of organic carbon in yellow paddy soilPLOS ONE

Dear Dr. wang,

Thank you for submitting your manuscript to PLOS ONE. After careful consideration, we feel that it has merit but does not fully meet PLOS ONE’s publication criteria as it currently stands. Therefore, we invite you to submit a revised version of the manuscript that addresses the points raised during the review process.

We look forward to receiving your revised manuscript.

Kind regards,

Dafeng Hui, Ph.D.

Academic Editor

PLOS ONE

**Journal Requirements:**

**Additional Editor Comments:**

I now have two reports from expert reviewers. Both reviewers recognized the merits of the study, but also raised some technique concerns. Please revise the manuscript based on the reviewers' comments and provide a detailed response to each concern.

Reviewers' comments:

Reviewer's Responses to Questions

**Comments to the Author**

1. Is the manuscript technically sound, and do the data support the conclusions?

Reviewer #1: Yes

Reviewer #2: Yes

2. Has the statistical analysis been performed appropriately and rigorously? 

Reviewer #1: Yes

Reviewer #2: N/A

3. Have the authors made all data underlying the findings in their manuscript fully available?

Reviewer #1: Yes

Reviewer #2: Yes

4. Is the manuscript presented in an intelligible fashion and written in standard English?

Reviewer #1: Yes

Reviewer #2: No

5. Review Comments to the Author

**Reviewer #1: **Comments

The research article by Fangchi Wang et al., titled “The impact of straw and its post-pyrolysis incorporation on functional microbes and mineralization of organic carbon in yellow paddy soil,” provides valuable insights into establishing a theoretical framework for the microecological mechanisms of soil carbon sequestration and informing rational fertilization practices for Guizhou's yellow paddy soils. The study is based on a five-year field experiment with four treatments: no fertilizer application (CK), chemical fertilizer only (NPK), straw combined with chemical fertilizer (NPKS), and biochar combined with chemical fertilizer (NPKB). By integrating indoor mineralization culture with metagenomic approaches, the authors analyzed the response of organic carbon mineralization and carbon cycle genes in typical paddy soil from Guizhou Province, China, to different fertilization treatments. The manuscript presents interesting results and could be accepted after necessary major revisions. Below, I provide specific comments to help improve the manuscript.

Specific comments:

Abstracts should provide a clear and concise summary of the study's purpose, methods, and key findings. Overly detailed or lengthy descriptions can detract from the immediate accessibility of the information. Summarizing the core outcomes more directly will help ensure the abstract captures readers' attention and communicates the research's significance effectively.

Other comments

Line 20: There is a typographical error with the use of 'The' in the middle. Please check for similar issues throughout the manuscript.

Present the percentage increase, decrease, or values of the indicators that were altered among the treatments.

What are the suggestions for future research based on your study? Please add this information at the end of the abstract.

Line 105-107: The stated objective of the study, which suggests providing a theoretical framework for the microecological mechanisms of soil carbon sequestration and informing fertilization practices, does not seem accurately aligned with the actual goals of the research. Could you please clarify the true objectives? What specific issues were you aiming to address? What practical outcomes did you hope to implement, and what problem are you trying to solve with this research?

Line 87: The sentence “Research indicates that the application of straw and biochar markedly influences genes associated with soil carbon fixation and degradation” add this sentence here “Additionally, combined straw return and other fertilizer significantly enhanced soil fertility and beneficial bacterial abundance, with soil organic carbon being the primary factor influencing bacterial community structure (Borny et al., 2024)” https://doi.org/10.56946/jspae.v3i1.404” and try to add recent citations in the introduction section.

Line 136: correct “phas e” remove the space.

Line 209-223, Please rewrite this section to improve the flow of the text. Avoid using the same terms repeatedly in consecutive sentences

Table 1: The statistical lettering seems incorrect. I recommend reanalyzing the table data or reviewing the analysis, as there may be errors in how it was reported. For example in DOC 226 is ab while 196 is a.

Many other mistakes are existed in the manuscript such as grammatical, extra spacing, etc which all should be addressed.

Regards Izhar Ali

**Reviewer #2: **Comments on PONE-D-24-40473

In this manuscript, a 30-day indoor soil mineralization culture test was conducted based on a five-year experimental field to investigate the effects of straw and biochar amendments on soil carbon mineralization and the functional genes of microbial communities. The study is relevant to soil management practices, particularly in the context of sustainable agriculture and carbon sequestration. However, it requires some revisions to improve clarity, depth, and coherence. Below are specific comments and suggestions.

Specific comments

1. Soil and experimental design: It is necessary to clarify whether the application of straw and biochar was conducted solely in the first year or if it was supplemented annually.

2. Could you please explain in detail why MBC were high under NPKS and NPKB treatments (Table 1), while the mineralization rate (Table 2, k) of NPKB showed no significant difference from the control?

3. The description of the treatments is clear, but consider summarizing them in a table format for quick reference.

4. Your discussion of the contrasting effects of straw and biochar on SOC mineralization is compelling. Highlight the long-term implications of these findings for soil management practices, particularly in terms of balancing immediate nutrient availability with long-term carbon sequestration.

5. Line287: It appears that Figure 2 does not clearly demonstrate that endochitinase (E3.2.1.14) exhibited the highest abundance under the NPKS treatment.

6. Line298-299: How does Figure 3 demonstrate that NPKS increased the gene abundance of MCEE, rbcL, and GAPDH?

7. Line 321: ‘MOO375’ should be amended to ‘M00375’. What do red and blue represent, respectively?

8. Line 339: Could you provide evidence that the carbon sequestration pathway is significantly positively correlated with pH? I cannot find it directly.

9. Line 346: It is more appropriate to modify (a) and (b) to ‘Abundance of Carbon Degrading Bacteria’ and ‘Abundance of Carbon Fixing Bacteria’.

10. Line 374: When using terms like ‘soil active organic carbon’, consider providing brief definitions or examples to ensure clarity. For instance, explain what constitutes active organic carbon in practical terms and how they differ from labile carbon.

6. PLOS authors have the option to publish the peer review history of their article (what does this mean?). If published, this will include your full peer review and any attached files.

Reviewer #1: No

Reviewer #2: No

---

## [Author Response · Author response to Decision Letter 0]

7 Nov 2024

Dear Reviewers,

Greetings!

I am Fangchi Wang from the College of Agriculture, Guizhou University, China. First and foremost, please allow me to extend our deepest appreciation to both of you and the other reviewer for taking the time out of your valuable schedules to review our manuscript titled "The impact of straw and its post-pyrolysis incorporation on functional microbes and mineralization of organic carbon in yellow paddy soil," and for offering your valuable feedback and suggestions. We hold your professional dedication and meticulous review in the highest regard.

After carefully considering the comments from Reviewer 1 and Reviewer 2, we have made thoughtful revisions to our manuscript. Here are some of the main changes we have made:

Reviewer #1: Comments

The research article by Fangchi Wang et al., titled “The impact of straw and its post-pyrolysis incorporation on functional microbes and mineralization of organic carbon in yellow paddy soil,” provides valuable insights into establishing a theoretical framework for the microecological mechanisms of soil carbon sequestration and informing rational fertilization practices for Guizhou's yellow paddy soils. The study is based on a five-year field experiment with four treatments: no fertilizer application (CK), chemical fertilizer only (NPK), straw combined with chemical fertilizer (NPKS), and biochar combined with chemical fertilizer (NPKB). By integrating indoor mineralization culture with metagenomic approaches, the authors analyzed the response of organic carbon mineralization and carbon cycle genes in typical paddy soil from Guizhou Province, China, to different fertilization treatments. The manuscript presents interesting results and could be accepted after necessary major revisions. Below, I provide specific comments to help improve the manuscript.

Specific comments:

1. Abstracts should provide a clear and concise summary of the study's purpose, methods, and key findings. Overly detailed or lengthy descriptions can detract from the immediate accessibility of the information. Summarizing the core outcomes more directly will help ensure the abstract captures readers' attention and communicates the research's significance effectively.

2. Line 20: There is a typographical error with the use of 'The' in the middle. Please check for similar issues throughout the manuscript.

3. Present the percentage increase, decrease, or values of the indicators that were altered among the treatments.

4. What are the suggestions for future research based on your study? Please add this information at the end of the abstract.

5. Line 105-107: The stated objective of the study, which suggests providing a theoretical framework for the microecological mechanisms of soil carbon sequestration and informing fertilization practices, does not seem accurately aligned with the actual goals of the research. Could you please clarify the true objectives? What specific issues were you aiming to address? What practical outcomes did you hope to implement, and what problem are you trying to solve with this research?

6. Line 87: The sentence “Research indicates that the application of straw and biochar markedly influences genes associated with soil carbon fixation and degradation” add this sentence here “Additionally, combined straw return and other fertilizer significantly enhanced soil fertility and beneficial bacterial abundance, with soil organic carbon being the primary factor influencing bacterial community structure (Borny et al., 2024)” https://doi.org/10.56946/jspae.v3i1.404” and try to add recent citations in the introduction section.

7. Line 136: correct “phas e” remove the space.

8. Line 209-223, Please rewrite this section to improve the flow of the text. Avoid using the same terms repeatedly in consecutive sentences.

9. Table 1: The statistical lettering seems incorrect. I recommend reanalyzing the table data or reviewing the analysis, as there may be errors in how it was reported. For example in DOC 226 is ab while 196 is a.

10. Many other mistakes are existed in the manuscript such as grammatical, extra spacing, etc which all should be addressed.

Responses to Reviewer 1:

1 Abstract Revision

We have refined the abstract to more directly highlight the purpose, methods, and main findings of our research, thereby enhancing the immediate accessibility of the information. The abstract now provides a clearer summary of our study and emphasizes its impact on soil carbon mineralization and the function of carbon cycle genes. The specific revisions are as follows (lines 17-31):

The impact of straw and biochar on carbon mineralization and the function of carbon cycle genes in paddy soil is important for soil nutrient management and the transformation of carbon pools. This research is based on a five-year field experiment with four treatments: no fertilizer application (CK); chemical fertilizer only (NPK); straw combined with chemical fertilizer (NPKS); and biochar combined with chemical fertilizer (NPKB). By integrating indoor mineralization culture with metagenomic approaches, we analyzed the response of organic carbon mineralization and carbon cycle genes in typical paddy soil from Guizhou Province, China, to different fertilization treatments. The result shows that the various fertilization treatments significantly increased the levels of soil organic carbon, dissolved organic carbon, microbial biomass carbon, and readily oxidizable organic carbon. The NPKS treatment increased the rate of soil organic carbon mineralization, whereas the NPKB treatment decreased it. Overall, the NPK and NPKB treatments increased the relative abundance of carbon fixation genes. The NPKS treatment increased the relative abundance of carbon degradation genes. The NPKS treatment increased the abundance of Proteobacteria, whereas the NPKB treatment decreased the abundance of Actinobacteria. Overall, Biochar can reduce carbon loss and enhance sequestration of soil carbon, whereas straw decreases soil organic carbon stability, accelerating the transformation of soil carbon pools. Future research should encompass long-term impact assessments to comprehensively understand the enduring effects of these fertilization treatments on soil carbon mineralization and the function of carbon cycle genes.

2 Suggestions for Future Research

Following your suggestion, we have added recommendations for future research at the end of the abstract to guide potential subsequent studies, as detailed below (lines 29-31):

Future research should encompass long-term impact assessments to comprehensively understand the enduring effects of these fertilization treatments on soil carbon mineralization and the function of carbon cycle genes.

3 Correction of Typographical Errors

We have corrected the typographical error on line 20 and thoroughly checked the entire manuscript to ensure the accuracy of all text, correcting any grammatical errors and formatting issues.

4 Display of Percentage Changes in Indicators

We have revised the manuscript to show the percentage increase, decrease, or value of indicators changed between treatments.

5 Recent Literature Cited

In response to the comments on lines 50-51, we have added citations to the most recent studies and included the latest literature in the introduction to provide an up-to-date background for our research. The specific citations are as follows (lines50-51, lines 89-91)：

[4] Raihan A, Chandra V L, Babla M, Shoaibur R M, Rashed Z M (2023) Economy-Energy-Environment Nexus: The Potential of Agricultural Value-Added Toward Achieving China’S Dream of Carbon Neutrality. Carbon Research. 2(1). https://doi.org/10.1016/j.igd.2024.100139

[5] Raihan A, Bari A B M M (2024) Energy-Economy-Environment Nexus in China: The Role of Renewable Energies Toward Carbon Neutrality. Innovation and Green Development. 3(3):100139-. https://doi.org/10.1007/s44246-023-00077-x

[18] Borny, N. R., Mostakim, G. M., Raihan, A., & Rahman, M. S. (2024). Synergistic Effects of Rice Straw Return and Nitrogen Fertilizer on Rhizosphere Bacterial Communities and Soil Fertility. Journal of Soil, Plant and Environment, 3(1), 41–58. https://doi.org/10.56946/jspae.v3i1.404

6 Clarification of Research Objectives

We have revisited and clarified our research objectives to ensure they align with the actual content and results of our study. Our objectives are as follows (lines 108-112):

This study is designed to evaluate the long-term effects of straw and biochar amendments on soil carbon cycling, with the objective of identifying fertilization strategies that can enhance soil carbon sequestration and reduce carbon emissions. Our aim is to determine the optimal agricultural management practices that not only improve soil quality and boost agricultural productivity but also contribute to the challenges of global climate change mitigation. 

7 Improvement in Textual Fluency and Accuracy of Statistical Data

Regarding the concerns raised on line 87 and lines 226-241, we have re-analyzed the data in the tables to ensure the accuracy of our statistical reporting. The specific details are as follows (lines 217-232):

The soil organic carbon (SOC) levels were significantly elevated by 73% and 36% under the NPKS and NPKB treatments, respectively, compared with the CK treatment (p < 0.05, for all subsequent comparisons) (Table 2). The addition of straw and biochar to the soil led to a substantial increase in dissolved organic carbon (DOC) levels, with the NPKS and NPKB treatments raising DOC by 38% and 23%, respectively, compared to the control, yet no significant difference was observed between the two treatments. In the NPKS and NPKB treatments, the microbial biomass carbon (MBC) was significantly increased by 40% and 20% compared to the control, respectively. While readily oxidizable carbon (ROC) levels remained relatively consistent across all treatments, the NPKS treatment recorded the highest values, with the biochar-amended and chemical treatments following suit. The composition of active SOC components was affected by the fertilization methods, with the NPKS treatment exerting the most notable impact. This particular treatment led to a significant reduction in the DOC/SOC ratio by 19% when compared to the control, a change not seen with other treatments. Despite no significant variations in the MBC/SOC ratio among the treatments, the straw-enriched treatment had the lowest MBC/SOC ratio, with the biochar-amended and chemical treatments showing reductions of 18% and 12%, respectively, compared to the control. Moreover, the straw-enriched treatment caused a significant 35% decrease in the ROC/SOC ratio in comparison to the control. These findings suggest that specific fertilization strategies can alter the dynamics of SOC components, potentially influencing soil fertility and carbon sequestration potential.

Reviewer #2: Comments

In this manuscript, a 30-day indoor soil mineralization culture test was conducted based on a five-year experimental field to investigate the effects of straw and biochar amendments on soil carbon mineralization and the functional genes of microbial communities. The study is relevant to soil management practices, particularly in the context of sustainable agriculture and carbon sequestration. However, it requires some revisions to improve clarity, depth, and coherence. Below are specific comments and suggestions.

Specific comments

1. Soil and experimental design: It is necessary to clarify whether the application of straw and biochar was conducted solely in the first year or if it was supplemented annually.

2. Could you please explain in detail why MBC were high under NPKS and NPKB treatments (Table 1), while the mineralization rate (Table 2, k) of NPKB showed no significant difference from the control?

3. The description of the treatments is clear, but consider summarizing them in a table format for quick reference.

4. Your discussion of the contrasting effects of straw and biochar on SOC mineralization is compelling. Highlight the long-term implications of these findings for soil management practices, particularly in terms of balancing immediate nutrient availability with long-term carbon sequestration.

5. Line287: It appears that Figure 2 does not clearly demonstrate that endochitinase (E3.2.1.14) exhibited the highest abundance under the NPKS treatment.

6. Line298-299: How does Figure 3 demonstrate that NPKS increased the gene abundance of MCEE, rbcL, and GAPDH?

7. Line 321: ‘MOO375’ should be amended to ‘M00375’. What do red and blue represent, respectively?

8. Line 339: Could you provide evidence that the carbon sequestration pathway is significantly positively correlated with pH? I cannot find it directly.

9. Line 346: It is more appropriate to modify (a) and (b) to ‘Abundance of Carbon Degrading Bacteria’ and ‘Abundance of Carbon Fixing Bacteria’.

10. Line 374: When using terms like ‘soil active organic carbon’, consider providing brief definitions or examples to ensure clarity. For instance, explain what constitutes active organic carbon in practical terms and how they differ from labile carbon.

Responses to Reviewer 2:

1 Soil and Experimental Design

We have clarified in lines 123-125 that the experiment commenced in 2019 for a five-year period, with the same quantities of straw, biochar, and chemical fertilizer being applied each year.

2 Proof of Gene Abundance in Figure 3

For lines 309-310, we clarified that the abundance of rbcl genes did not increase; the color changes in the figure (from blue to white to red) indicate gene abundance from low to high.

3 Explanation of Endochitinase Abundance in Figure 2

In lines 311-312, we clarified that under NPKS treatment, the abundance of endochitinase (E3.2.1.14) was the highest, reaching 1.84%, while the control (CK), NPK, and NPKB treatments were 1.74%, 1.76%, and 1.81%, respectively.

4 Code Correction and Legend Color Representation

We have corrected "MOO375" to "M00375" and explained the specific meanings of red and blue in the legend on line 358.

5 Correlation between Carbon Sequestration Pathways and pH Value

For line 358, we apologize for the input error and clarify that there is no significant positive correlation between carbon sequestration pathways and pH value.

6 Explanation for High MBC Content in NPKS and NPKB Treatments

In lines 396-407, we have detailed the reasons for the significant increase in microbial biomass carbon (MBC) in NPKS and NPKB treatments, and discussed the possible factors for the non-significant difference in soil organic carbon turnover rate (k value) between NPKB treatment and the control.

However, while the NPKS and NPKB treatments significantly increased microbial biomass carbon (MBC), we observed no significant difference in the rate of soil organic carbon turnover (k value) between the NPKB treatment and the control (Table 3). This unexpected outcome may be attributed to several factors. Firstly, under the NPKS treatment, the decomposition of straw rapidly releases nutrients that can be quickly utilized by microorganisms, thereby increasing microbial biomass[35]. In contrast, biochar in the NPKB treatment, due to its high specific surface area and porosity, effectively adsorbs and retains organic carbon and nutrients in the soil, reducing their loss. Moreover, the high chemical stability of biochar makes it less susceptible to microbial decomposition, which may slow the turnover rate of soil organic carbon, resulting in no significant increase in the carbon turnover rate (k value) even with high MBC[36]. Additionally, the incorporation of biochar may alter the structure of the soil microbial community, potentially promoting the growth of microorganisms with slower decomposition rates, thus lowering the overall mineralization rate. These findings indicate that the impact of different organic materials on soil carbon cycling is complex and influenced by multiple factors[37]. 

7 Summary of Treatment Methods

We have summarized the treatment methods in Table 1 for quick reference.

Table 1 Fertilization amount of different treatments//kg•hm-2

Treatment Nitrogen (N) (kg/hm²) Phosphorus (P₂O₅) (kg/hm²) Potassium (K₂O) (kg/hm²) Straw (t/hm²) Biochar (t/hm²)

CK 0 0 0 0 0

NPK 150 148 230 0 0

NPKS 150 148 230 15 0

---

## [Editor Report · Decision Letter 1]

20 Nov 2024

The impact of straw and its post-pyrolysis incorporation on functional microbes and mineralization of organic carbon in yellow paddy soil

PONE-D-24-40473R1

Dear Dr. wang,

We’re pleased to inform you that your manuscript has been judged scientifically suitable for publication and will be formally accepted for publication once it meets all outstanding technical requirements.

Kind regards,

Dafeng Hui, Ph.D.

Academic Editor

PLOS ONE

Additional Editor Comments (optional):

The authors made great efforts and addressed the reviewers' concerns. I recommend Accept.

Minor changes:

L28: delete "Overall,"

L223: delete "and analysis"

L535-539: please use past tense to describe the results here. For example, change are to were, activates to activated, alters to altered, accelerates to accelerated, does to did, elevates to elevated.
---

## [Editor Report · Acceptance letter]

11 Dec 2024

PONE-D-24-40473R1 

PLOS ONE

Dear Dr. wang, 

I'm pleased to inform you that your manuscript has been deemed suitable for publication in PLOS ONE. Congratulations! Your manuscript is now being handed over to our production team.

Kind regards, 

on behalf of

Dr. Dafeng Hui 

Academic Editor

PLOS ONE